# Dietary cholesterol reduces blood pressure and alters lipid profiles in stroke-prone spontaneously hypertensive rats

Yutaro Nishikata[1], Kenjiro Tatematsu[1]*, Yoshiaki Saito[2], Toshiyuki Matsunaga[1], Naoki Ohara[3]

**1** Department of Bioinformatics, Gifu Pharmaceutical University, Gifu, Japan, **2** Department of Pathology, Food and Drug Safety Center, Hatano Research Institute, Kanagawa, Japan, **3** Open Research Center for Lipid Nutrition, Kinjo Gakuin University College of Pharmacy, Aichi, Japan

* tateken@gifu-pu.ac.jp

## Abstract

Although cholesterol (Chol) is widely recognized as a risk factor for cardiovascular disease, dietary Chol intake has been reported to extend the lifespan of stroke-prone spontaneously hypertensive rats (SHRSP). The mechanisms responsible for this paradoxical effect remain unclear. The present study examined changes in organ lipid profiles and associated molecular factors in SHRSP rats fed a Chol-enriched diet. Four-week-old male SHRSP/Izm rats were assigned to three groups and fed ad libitum for 12 weeks with either a control diet (Ctr), a diet supplemented with 1% w/w Chol (Chol), or a diet containing 1% w/w Chol plus 0.025% w/w lovastatin to suppress endogenous Chol synthesis. Systolic blood pressure was measured before and after the feeding period, and tissues were collected for analyses of sterol content, fatty acid composition, prostaglandin $E_2$ ($PGE_2$) levels, and renal histopathology. Relative to the Ctr group, the Chol group exhibited a significant 9–10% reduction in systolic blood pressure. This reduction was accompanied by pronounced alterations in lipid profiles, including changes in phytosterol content and decreased arachidonic acid ratios in serum and kidney. There was a downward trend in hepatic $PGE_2$ levels, and a similar tendency was observed in the kidney. Comparable changes in lipid profiles were observed in the Chol + lovastatin group. Histological analysis revealed modest attenuation of renal pathological features in Chol-fed rats. This study demonstrates for the first time that dietary Chol reduces renal phytosterol accumulation and suppresses the AA-$PGE_2$ axis, changes that coincide with a 9–10% reduction in systolic blood pressure and attenuated glomerular inflammation. These integrated findings provide a mechanistic framework linking dietary Chol to the previously reported lifespan extension in this stroke-prone model. Although these changes may contribute to improved renal pathology, further studies are required to clarify causal relationships.

**Data availability statement:** All relevant data are within the manuscript and its Supporting Information files.

**Funding:** This work was supported by JSPS KAKENHI Grant Numbers JP22K05526. The funders had no role in study design, data collection and analysis, decision to publish, or preparation of the manuscript.

**Competing interests:** The authors have declared that no competing interests exist.

## Introduction

Sterols and fatty acids (FAs) are prevalent dietary lipids that serve essential physiological functions. Cholesterol (Chol), which is predominantly obtained from animal-derived foods, constitutes a fundamental component of animal cell membranes and functions as a precursor for steroid hormones and bile acids. Elevated total Chol levels are a well-established risk factor for atherosclerotic cardiovascular disease because they promote the development of Chol-rich plaques, although LDL-Chol is considered a more direct causal factor in atherosclerosis [1]. In contrast, epidemiological studies and meta-analyses have consistently demonstrated an inverse association between serum total Chol levels and the risk of hemorrhagic stroke [2]. The mechanisms underlying this paradoxical association remain insufficiently characterized.

Stroke-prone spontaneously hypertensive rats (SHRSP) have been established as a model for human hypertension and hemorrhagic stroke [3]. In this strain, lifespan is curtailed by fatal strokes secondary to severe hypertension and renal impairment [4]. Notably, SHRSP rats exhibit a paradoxical response to dietary lipids. Diets rich in soybean- or rapeseed-derived phytosterols (PhyS), which mainly contain β-Sitosterol (Sito), shorten survival by raising blood pressure [5–7], whereas a Chol-rich diet prolongs lifespan [8]. This strain is characterized by a mutation in the ATP-binding cassette transporter G5 (ABCG5), a sterol transporter involved in Chol efflux, resulting in accelerated PhyS accumulation in the vasculature [9]. The accumulated PhyS can partially replace membrane Chol, which may reduce membrane integrity and increase vascular fragility [10]. Under the severe hypertension characteristic of SHRSP rats, these fragile membranes, particularly in the brain, may predispose cerebral vessels to rupture, contributing to hemorrhagic stroke [11].

Chol and PhyS have been found to share similar intestinal absorption pathways. Both sterols have been observed to form micelles with bile acids and are taken up into epithelial cells via Niemann-Pick C1-like 1 (NPC1L1) [12]. After this process, approximately 50% of Chol and 90% of PhyS are re-excreted into the small intestinal lumen by the ATP-binding cassette transporter G5/8 (ABCG5/8) [12]. Consequently, PhyS exerts a competitive inhibition on Chol absorption, thereby demonstrating a Chol-lowering effect in the blood. Conversely, statins, which function as inhibitors of 3-hydroxy-3-methylglutaryl-CoA (HMG-CoA) reductase (HMGCR), represent the most prevalent treatment modality for hypercholesterolemia in clinical practice [13]. In addition to lowering Chol, statins exert several pleiotropic effects, including attenuation of oxidative stress, suppression of vascular inflammation, and improvement of blood pressure regulation [14–16]. Although statins have been reported to extend survival and reduce stroke volume in SHRSP rats [17], their effects in the context of a Chol-enriched diet have not been fully examined.

Chol and FA metabolism is closely interconnected, and dietary Chol influences FA metabolism through transcriptional regulators such as sterol regulatory element-binding proteins 1c (SREBP-1c) [18]. Polyunsaturated FAs (PUFAs), specifically those belonging to the n-6 and n-3 series, play key roles in the regulation of inflammation [19]. Specifically, arachidonic acid (AA) (20:4n-6) is metabolized into

prostaglandins (PGs) and other compounds that regulate inflammatory responses [20]. Therefore, alterations in sterol composition may exert an influence on FA-derived inflammatory pathways, potentially by altering the substrate availability through the regulation of FA metabolic enzymes. However, the mechanisms by which dietary Chol intake modifies sterol composition and associated FA metabolic pathways in SHRSP rats remain to be elucidated, as do the relationships between these changes and the pathological features characteristic of this strain. Chol and PhyS are ingested daily, and statins are frequently utilized in clinical settings. Consequently, clarifying the underlying mechanisms through which these interactions may contribute to disease-relevant pathways in cerebrovascular disease models may provide insights from both nutritional and clinical perspectives.

The objective of this study was to elucidate the impact of dietary Chol and statin on sterol and FA metabolism in SHRSP rats, and to determine the role of these metabolic changes in the pathological characteristics that are hallmarks of this strain.

## Materials and methods

### Test diets

The basal diet was LabDiet 5001 Laboratory Rodent Diet in meal form. According to the datasheet provided by the manufacturer, the chemical composition of the diet includes 24.1% protein, 5.1% fat (ether extract), 5.3% crude fiber and 2.86 kcal/g of metabolizable energy. The calories are provided as follows: 28.9% from protein, 13.6% from fat and 57.5% from carbohydrates. The experiment involved the preparation of three test diets: a control (Ctr) group, a Chol group, and a Chol + Statin group. Through the experiment, the test diets were provided in meal form. The experimental diets for the Chol and Chol + Statin groups were prepared by mixing the basal diet with 1% w/w Chol powder (FUJIFILM Wako Pure Chemical Co., Osaka, Japan) or with 1% w/w Chol powder and 0.025% w/w lovastatin powder (Tokyo Chemical Industry Co., Tokyo, Japan), respectively. The diets were prepared using a mixer (PRO-GMS5; KIPROSTAR, Gifu, Japan) for over 30 minutes. They were then collected from two different places for analysis using gas chromatography to confirm homogeneity of the diets. The diets were stored at temperatures below −20°C until use. Tables 1 and 2 present the FA compositions and sterol contents.

### Animals

The present study was conducted in strict accordance with the recommendations set forth in the Guide for the Care and Use of Laboratory Animals of the National Institutes of Health [21]. The study protocol was approved by the Institutional Animal Care and Use Committee of Gifu Pharmaceutical University (Gifu, Japan; Permit No. 2023−058) and the Committee for Animal Research and Welfare of Gifu University (Permit No. 2023−046). Terminal procedures were performed under deep anesthesia, followed by euthanasia, and all efforts were made to minimize animal suffering. All rats were maintained under specific pathogen-free conditions at 23°C ± 3°C and 55% ± 15% humidity in a 12:12-h light/dark cycle.

Eighteen male SHRSP/Izm rats, aged four weeks, were procured from Japan SLC Inc. (Shizuoka, Japan). Following acclimatization, the animals were randomly assigned to three groups (Ctr, Chol, and Chol + Statin; n = 6 per group) and housed in groups of three per cage. Animals in each group were permitted unrestricted access to the test diet and drinking water for a period of 12 weeks. Body weight and food consumption were monitored on a weekly basis. The subjects' food intake was meticulously monitored using a specialized feeding device. Furthermore, blood pressure was measured before and after the feeding period using a tail-cuff method on a blood pressure monitor (MK-2000ST, Muromachi Kikai Co., Tokyo, Japan). At the conclusion of the feeding period, the animals were anesthetized by inhalation of isoflurane. Following euthanasia, the major organs were promptly harvested, weighed, frozen in liquid nitrogen, and stored at −80°C until use. Also, we collected feces from each cage at the end of the feeding period, and stored at −80°C until use.

 

**Table 1. FA compositions and total levels of the test diets.**

| FA(%) | Ctr | Chol | Chol+Statin |
|---|---|---|---|
| 14:0 | 1.5 | 1.5 | 1.5 |
| 16:0 | 21.9 | 22.0 | 21.9 |
| 16:1 | 2.1 | 2.3 | 2.2 |
| 18:0 | 7.6 | 7.5 | 7.7 |
| 18:1n-9 | 30.4 | 30.5 | 30.2 |
| 18:1n-7 | 1.5 | 1.4 | 1.6 |
| 18:2n-6 | 30.4 | 30.1 | 30.1 |
| 18:3n-6 | 0.1 | 0.1 | 0.1 |
| 18:3n-3 | 2.4 | 2.3 | 2.4 |
| 20:0 | 0.5 | 0.5 | 0.5 |
| 20:1n-9 | 0.1 | 0.1 | 0.1 |
| 20:5n-3 | 0.8 | 0.8 | 0.9 |
| 22:0 | 0.1 | 0.1 | 0.1 |
| 22:6n-3 | 0.6 | 0.7 | 0.8 |
| 24:0 | 0.1 | 0.1 | 0.1 |
| Total FA (mg/g diet) | 44.5 | 44.5 | 43.0 |

Fatty acids (FA) in each diet were derived from the control diet (5001 Laboratory Rodent Diet Meal). Values are presented as means (n = 2). Abbreviations: Ctr, control group; Chol, cholesterol group; Chol+Statin, cholesterol+lovastatin group.

**Table 2. Sterol levels of the test diets.**

| Sterol (mg/g diet) | Ctr | Chol | Chol+Statin |
|---|---|---|---|
| Cholesterol | 0.293 | 10.599 | 10.394 |
| Campesterol | 0.075 | 0.081 | 0.083 |
| Stigmasterol | 0.028 | 0.031 | 0.032 |
| β-Sitosterol | 0.285 | 0.283 | 0.289 |
| Total phytosterol | 0.388 | 0.395 | 0.404 |

The Chol and Chol+Statin diets were supplemented with 1% w/w Chol. All phytosterols in each diet were derived from the control diet (5001 Laboratory Rodent Diet Meal). Values are presented as means (n = 2). Abbreviations: Ctr, control group; Chol, cholesterol group; Chol+Statin, cholesterol+lovastatin group.

## Lipid analysis

The composition of the test diet, serum, liver, and kidney in terms of FA and sterol was determined according to the method described in our previous report, with certain modifications [6]. In brief, total lipids were extracted using the Bligh and Dyer method [22], and FAs were converted to methyl esters using a 10% HCl-methanol solution. (Tokyo Chemical Industry Co.). Following the extraction of methyl esters by means of petroleum ether, gas chromatography (GC-2010; Shimadzu Co., Kyoto, Japan), equipped with a DB-225 capillary column (Agilent Technologies, Inc., Santa Clara, CA), for the analysis of FAs. Heptadecanoic acid was utilized as the internal standard. The composition of FA in the kidney was measured after the separation of phospholipids (PL) and neutral lipids, including free FA, by solid-phase extraction using the NH2 column (Bond Elut NH2; Agilent Technologies, Inc.).

For the purpose of sterol analysis, each tissue specimen was subjected to an incubation process with a 20% potassium hydroxide ethanol solution at a temperature of 100°C for a duration of two hours. This process, known as saponification, involved the use of betulin as an internal standard. The sterol fraction was extracted with hexane and converted to a trimethylsilyl derivative using TMS-HT® reagent (Tokyo Chemical Industry Co.). The quantification of these levels was performed by gas chromatography (GC) using a capillary column (DB-1; Agilent Technologies, Inc.).

## Quantitative analysis of mRNA expression

Total RNA was extracted from liver tissue using TRIzol™ reagent (Thermo Fisher Scientific Inc., Madison, WI) according to the manufacturer's instructions. RNA samples from each group (n = 6) were then pooled and treated with DNase I (Promega Corporation, Madison, WI). Real-time RT-PCR was performed using the RNA-direct® SYBR™ Green Realtime PCR Master Mix (Toyobo Co., Osaka, Japan) on a StepOne Real-Time PCR System (Thermo Fisher Scientific Inc.). Each pooled RNA sample was analyzed in duplicate, and the experiment was independently repeated three times. Gene expression was calculated using the ΔΔCt method and normalized to β-actin (housekeeping gene). The primer sequences utilized for RT-PCR are enumerated in Table 3. Although primer efficiencies were not explicitly determined, all primers showed single peaks in melting curve analysis, indicating specific amplification.

## Biochemical assay

The concentrations of prostaglandin $E_2$ ($PGE_2$) in the liver and kidneys were evaluated using the $PGE_2$ Monoclonal ELISA Kit (Cayman Chemical Co., Ann Arbor, MI).

**Table 3. The primer sequences used for RT-PCR.**

| Target | Forward (5'→3') | Reverse (5'→3') |
| --- | --- | --- |
| β-actin | cccgcgagtacaaccttct | cgtcatccatggcgaact |
| Abcg5 | gaatgtgtccttcagcgtca | gctggcatgatttgatgttc |
| Abcg8 | cgtcagatttccaatgacttcc | tccgtcctccagttcatagtac |
| Npc1l1 | ttgaggtcttcccttacacgatctcca | aggtgggcacaaagcaaagagcaa |
| Hmgcr | gacctttctagagcgagtgcat | gctatattctcccttacttcat |
| Ldlr | tgctactggccaaggacat | ctgggtggtcggtacagtg |
| Srebf1c | cgcttcttacagcacagcaa | tgcccaaggacaagcgccta |
| Scd1 | gaagcgagcaaccgacag | ggtggtcgtgtaggaactgg |
| Lxra | agggctccaggaagagatgt | caactccgttgcagagtcag |
| Elovl5 | tcgatgcgtcactcagtacc | cctttgactcgtgtgtctcg |
| Fads1 | aactggtttgtgtggggtgacg | gagacccagtccacattccg |
| Fads2 | ggcacttaaacggtgcgtcc | tgcaggctctttatgtcggg |
| Cox-2 | ccagagcagagagatgaaatacca | gcaggcgggatacagttc |
| Ep3 | caacctggcgaccatcaaag | aagctggatagccgtctccg |
| Ep4 | acaccacctcgctgagaact | gctcccactaacctcatcca |
| Sod1 | tgtgtccattgaagaatcgtgtga | tcttgtttctcgtggaccacc |

Abbreviations: *Abcg5*, ATP-binding cassette sub-family G member 5; *Abcg8*, ATP-binding cassette sub-family G member 8; *Npc1l1*, Niemann-Pick C1-like 1; *Hmgcr*, 3-hydroxy-3-methyl-glutaryl-coenzyme A (HMG-CoA) reductase; *Ldlr*, Low density lipoprotein receptor, *Srebf1c*, Sterol regulatory element-binding transcription factor 1c; *Scd1*, Stearoyl-CoA desaturase 1; *Lxra*, Liver X receptor α; *Elovl5*, Elongation of very long chain fatty acids 5; *Fads1*, Fatty acid desaturase 1; *Fads2*, Fatty acid desaturase 2; *Cox-2*, Cyclooxygenase-2; *Ep3*, Prostaglandin $E_2$ receptor 3, *Ep4*, Prostaglandin $E_2$ receptor 4; *Sod1*, Superoxide dismutase 1.

The activity of superoxide dismutase (SOD) in serum, liver, and kidney was measured using the SOD Assay Kit – WST (Dojin Molecular Technology Co., Kumamoto Prefecture) after pooling each tissue. In terms of serum, the hemolysis checks were not performed. All procedures were performed in strict accordance with the instructions provided with the kits.

## Histological analysis

Kidney tissue samples were fixed using 10% formalin neutral buffer solution and embedded in paraffin. The samples were then cut into 4-μm horizontal sections and stained with hematoxylin and eosin for histological examination. The process of embedding and staining was subcontracted to Morphotechnology Co. (Hokkaido, Japan). A pathologist then compared each sample with normal tissue in a blinded manner and assigned scores using a five-grade scoring system: – (no abnormal change), ± (very slight change), + (slight change), ╫ (moderate change), and ╫╫ (marked change). A grade of ╫╫ was assigned when lesions involved more than 80% of the examined area, with other grades distributed accordingly. In this evaluation, spontaneous lesions common in rodents, such as chronic progressive nephropathy, were also taken into account. For each animal (n = 6 per group), one section from bilateral kidneys was prepared, and the entire field of each section was examined to ensure a representative assessment.

## Statistical analysis

Statistical analysis was performed using KyPlot v.6.0 software (Keyence, Osaka, Japan). All data are expressed as the mean ± standard deviation (SD). Due to the small sample size (n = 6), non-parametric analyses were performed. Group differences were assessed using the Kruskal–Wallis test, followed by the Steel–Dwass test for multiple comparisons. No outlier exclusion was performed, and all data points were included in the analysis. In all cases, the significance level was set at $p < 0.05$. Statistical analysis was not performed on the ordinal histological data due to the low frequency and subtle nature of the observed changes, which precluded meaningful quantitative comparison.

## Results

### Body weight, food intake, blood pressure, and organ weight

Body weight and food intake were monitored weekly over the 12-week feeding period, and group differences were evaluated at each time point; no significant differences were observed (Fig 1A and 1B). Following the 12-week feeding period, a significant decrease in systolic blood pressure (SBP) was observed in the Chol group (223 ± 6 mmHg) compared with the Ctr group (240 ± 11 mmHg) (Fig 2). In the Chol + Statin group, the mean SBP decreased to 220 ± 11 mmHg, a level comparable to that of the Chol group, although this change did not reach statistical significance relative to the Ctr group. Table 4 presents the organ weight at the time of sacrifice. Liver weight was significantly higher in the Chol and Chol + Statin groups than in the Ctr group. However, no significant differences were observed in the weights of the other organs.

### Lipid analysis

An examination of sterol absolute concentrations and FA absolute concentrations and relative compositions in the liver, serum, and kidney was conducted to ascertain the effects of dietary Chol and lovastatin in SHRSP rats. Hepatic Chol levels were increased in both the Chol and Chol + Statin groups compared with the Ctr group, with no significant difference between these two groups (Fig 3A). Similarly, serum Chol levels were increased in both the Chol and Chol + Statin groups relative to the Ctr group (Fig 3B). In contrast, renal Chol levels did not significantly differ among the three diet groups (Fig 3C). Regarding PhyS, hepatic campesterol (Camp) and total PhyS levels were increased in both the Chol and Chol + Statin groups, whereas total PhyS levels were decreased in both serum and kidney (Fig 3A-3C). A comparative analysis revealed a decrease in β-sitosterol (Sito) absolute concentrations across all the tissues examined within the Chol

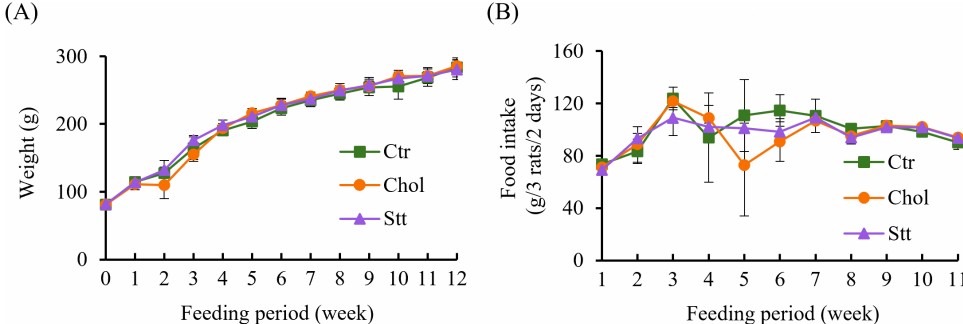

**Fig 1. Body weight and food intake of the SHRSP rats. (A)** Body weight and **(B)** food intake during the experimental period. Values are presented as mean ± SD (n = 6 rats/group for body weight; n = 2 cages/group for food intake). Error bars represent SD. No significant differences were observed among the three groups. Abbreviations: Ctr, control group; Chol, cholesterol group; Chol + Statin, cholesterol + lovastatin group; SD, standard deviation; SHRSP, stroke-prone spontaneously hypertensive rats.

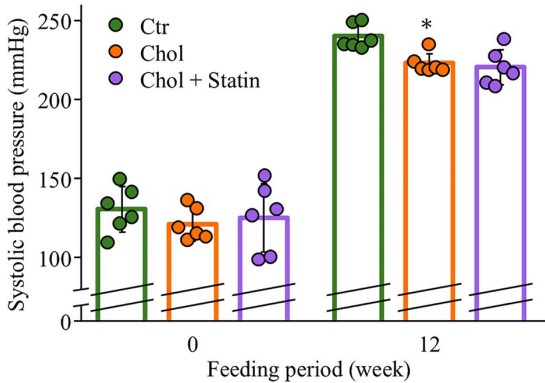

**Fig 2. Systolic blood pressure of the SHRSP rats.** Values are presented as mean ± SD ($n$ = 6/group). *$p$ < 0.05 vs. Ctr. Abbreviations: Ctr, control group; Chol, cholesterol group; Chol + Statin, cholesterol + lovastatin group; SD, standard deviation; SHRSP, stroke-prone spontaneously hypertensive rats.

group when contrasted with the Ctr group. Conversely, lovastatin supplementation led to a decrease in hepatic Camp and total PhyS levels relative to the Chol diet, while kidney sterol levels remained largely unaltered.

FA composition in the liver and serum was determined from total lipid extracts. As illustrated in Table 5, there was a significant increase in the total FA level (mg/g tissue) in the liver in both the Chol and Chol + Statin groups in comparison with the Ctr group. With respect to the composition of FA (%), an increase in palmitoleic acid (16:1n-7), oleic acid (18:1n-9), and linoleic acid (18:2n-6), accompanied by a decrease in palmitic acid (16:0), stearic acid (18:0), and AA, was observed in both the liver and serum of the Chol group in comparison with the Ctr group (Tables 5 and 6). The kidney analysis focused on the phospholipid (PL) fraction (isolated by column chromatography), as AA in PL is more closely associated with inflammatory processes. In the kidney PL, there was an increase of the composition (%) in 18:2n-6 and a decreasing tendency in AA (see Table 7). In the Chol + Statin group, AA composition (%) exhibited a decrease in serum, liver, and kidney PL compared with the Ctr group. Notably, among n-6 PUFAs, a consistent reduction tendency in AA composition (%) and absolute concentration (mg/g tissue) were observed across multiple tissues (Fig 4).

**Table 4. Organ weights of SHRSP rats.**

| Weight (g) | Ctr | Chol | Chol+Statin |
|---|---|---|---|
| Body | 284±14.2 | 286±8.97 | 280±14.6 |
| Liver | 9.07±0.83 | 11.27±0.66* | 11.35±1.20* |
| Kidney (Left) | 1.11±0.08 | 1.09±0.06 | 1.12±0.07 |
| Kidney (Right) | 1.07±0.11 | 1.09±0.07 | 1.09±0.06 |
| Brain | 1.86±0.03 | 1.82±0.08 | 1.79±0.07 |
| Organ/body weight (%) | Ctr | Chol | Chol+Statin |
| Liver | 3.20±0.29 | 3.94±0.23* | 4.05±0.43* |
| Kidney (Left) | 0.39±0.03 | 0.38±0.02 | 0.40±0.02 |
| Kidney (Right) | 0.38±0.04 | 0.38±0.02 | 0.39±0.02 |
| Brain | 0.65±0.01 | 0.64±0.03 | 0.64±0.03 |

Values are presented as mean±SD (n=6/group). *$p < 0.05$ vs. Ctr. Abbreviations: Ctr, control group; Chol, cholesterol group; Chol+Statin, cholesterol+lovastatin group; SD, standard deviation; SHRSP, stroke-prone spontaneously hypertensive rats.

## Biochemistry

As shown in Fig 5A and 5B, hepatic PGE$_2$ levels were significantly decreased in both the Chol and Chol+statin groups compared with the Ctr group (both $p<0.001$ vs. Ctr), whereas renal PGE$_2$ levels showed a non-significant decreasing trend (Chol vs. Ctr, $p=0.21$; Chol+Statin vs. Ctr, $p=0.14$).

S1 Fig shows that hepatic SOD activity was lower in both the Chol and Chol+Statin groups than in the Ctr group. Serum SOD activity was increased in the Chol group compared with the Ctr group, whereas there was no significant difference between the Ctr and Chol+Statin groups. Renal SOD activity was not significantly affected by either diet.

## mRNA expression

The hepatic mRNA expression levels are shown in Fig 6. In the liver, *Abcg5* and *Abcg8* are involved in bile excretion, while *Npc1l1* is involved in bile reabsorption [23]. The analysis revealed that the genes implicated in sterol transport (*Abcg5, Abcg8,* and *Npc1l1*) exhibited more than a twofold increase in the Chol group compared with the Ctr group. Conversely, these genes demonstrated a decrease in expression levels in the Chol+Statin group (Fig 6A). The expression of *Hmgcr* was found to be significantly reduced in both the Chol and Chol+Statin groups; however, a more pronounced decrease was observed in the Chol+Statin group. While *Ldlr* expression did not differ significantly across the groups. However, it tended to be lower in the Chol+Statin group than in the Chol group ($p=0.063$).

Among the genes implicated in FA metabolism, sterol regulatory-element binding factor 1c (*Srebf1c*) and stearoyl-CoA desaturase 1 (*Scd1*) exhibited increased expression levels in the Chol group (Fig 6B). In the Chol+Statin group, *Scd1* expression levels were lower than those observed in the Chol group; however, they remained higher than those seen in the Ctr group. Furthermore, the expression of liver X receptor α (*Lxra*) did not exhibit a significant difference between the Ctr and Chol groups; however, it demonstrated a significant decrease in the Chol+Statin group compared with the Ctr group. The expression of elongation of very long chain fatty acids 5 (*Elovl5*) was found to be decreased in the Chol group, whereas the expression levels of fatty acid desaturase 1 (*Fads1*) and *Fads2* remained constant. Conversely, *Fads1, Fads2,* and *Elovl5* exhibited a marked decrease in the Chol+Statin group.

With respect to the AA–PGE$_2$ pathway, the expression levels of cyclooxygenase 2 (*Cox-2*), prostaglandin E2 receptor 3 (*Ep3*), and Ep4 were increased in the Chol group compared with the Ctr group by approximately 1.8-, 1.4-, and 1.8-fold, respectively. Conversely, *Cox-2* and *Ep4* were decreased in the Chol+Statin group (Fig 6C). The expression of superoxide dismutase 1 (*Sod1*) was found to be reduced in both diet groups.

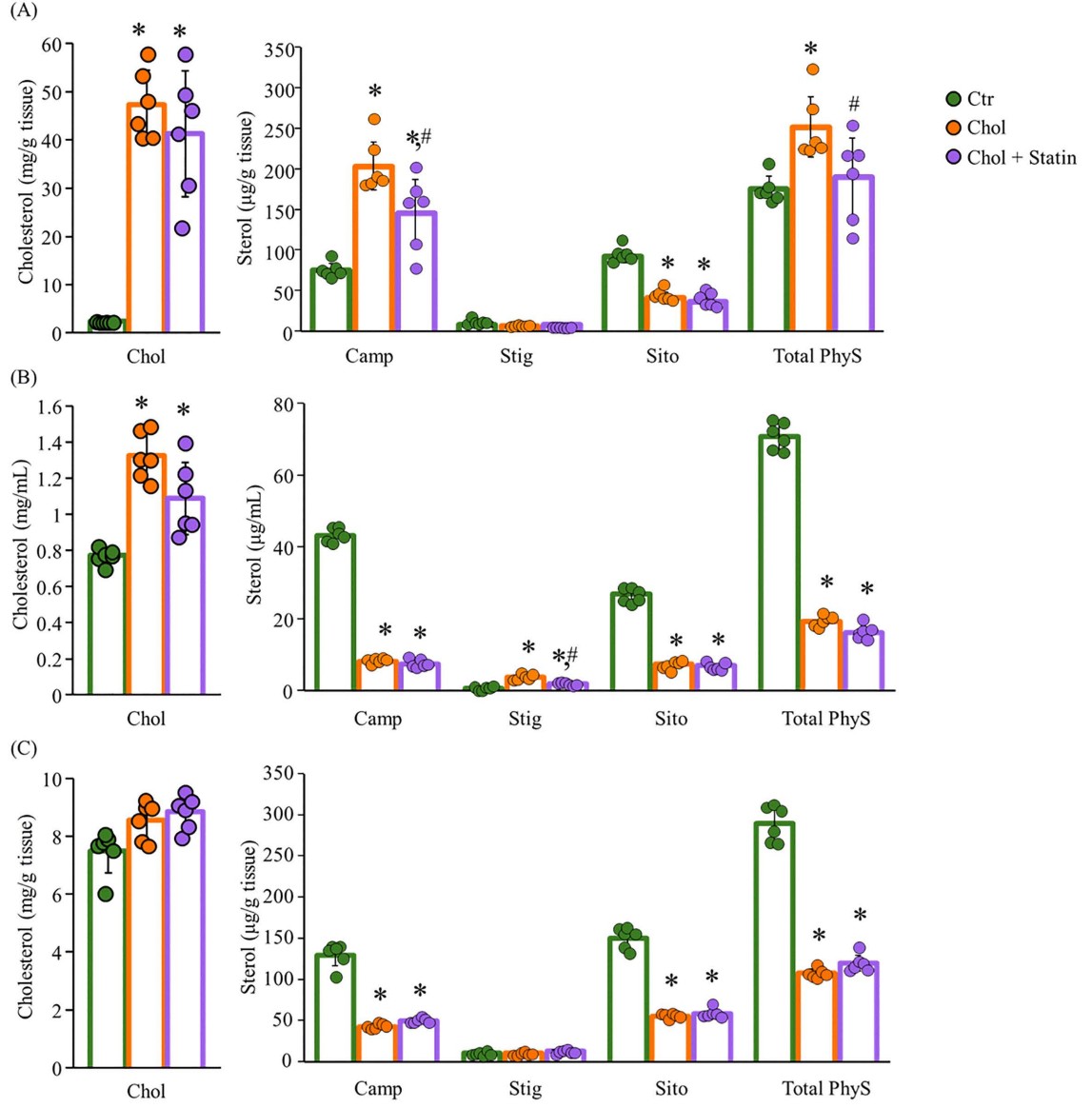

**Fig 3. Sterol levels in the liver, serum, and kidney of SHRSP rats. (A)** Hepatic, **(B)** serum and **(C)** renal Chol levels (left row) and PhyS levels (right row). Values are presented as mean ± SD ($n$ = 6/group). *$p$ < 0.05 vs. Ctr; #$p$ < 0.05 vs. Chol. Abbreviations: Ctr, control group; Chol, cholesterol group; Chol + Statin, cholesterol + lovastatin group; Camp, campesterol; Stig, stigmasterol; Sito, β-Sitosterol; PhyS, phytosterol; SD, standard deviation; SHRSP, stroke-prone spontaneously hypertensive rats.

## Histological analysis

The evaluation of kidney sections that had been stained with hematoxylin and eosin is illustrated in Fig 7 and detailed in Table 8. Slight infiltration of inflammatory cells in glomeruli (+) was observed in three of six rats in the Ctr group, whereas only very slight infiltration (±) was observed in the Chol and Chol + Statin groups. The presence of inflammatory cells in the interstitium (+) was observed in one rat in the Ctr group. However, basophilia in the tubule and cortex (+) was observed in one or two rats in each group, and the grades of granular cast and hyaline cast in the tubule did not exceed very slight (±) in any group.

**Table 5. FA compositions and levels in the liver of SHRSP rats.**

| FA (%) | Ctr | Chol | Chol+Statin |
|---|---|---|---|
| 14:0 | 0.3±0.1 | 0.3±0.0 | 0.3±0.0 |
| 16:0 | 22.1±1.1 | 17.6±0.4[*] | 17.7±0.6[*] |
| 16:1 | 0.7±0.2 | 3.7±0.5[*] | 3.2±1.0[*] |
| 18:0 | 21.0±1.0 | 8.6±1.3[*] | 9.4±2.3[*] |
| 18:1n-9 | 8.8±0.9 | 31.1±0.7[*] | 28.7±3.9[*] |
| 18:1n-7 | 2.5±0.1 | 2.8±0.1 | 2.6±0.5 |
| 18:2n-6 | 15.8±1.4 | 21.9±0.5[*] | 22.5±1.0 |
| 18:3n-6 | 0.3±0.0 | 0.2±0.0 | 0.2±0.0 |
| 18:3n-3 | 0.4±0.0 | 0.9±0.1[*] | 0.9±0.1[*] |
| 20:0 | tr. | 0.2±0.0[*] | 0.2±0.0[*] |
| 20:1n-9 | tr. | 0.2±0.0[*] | 0.2±0.0 |
| 20:2n-6 | tr. | tr. | tr. |
| 20:3n-9 | 0.2±0.0 | 0.3±0.0[*] | 0.3±0.1[*] |
| 20:3n-6 | tr. | tr. | tr. |
| 20:4n-6 | 19.4±4.5 | 7.5±0.5[*] | 8.9±1.7[*] |
| 20:5n-3 | 0.6±0.1 | 0.7±0.0 | 0.6±0.1 |
| 22:5n-3 | 1.2±0.1 | 0.9±0.1[*] | 0.9±0.1[*] |
| 22:6n-3 | 6.3±0.4 | 3.0±0.1[*] | 3.3±0.4[*] |
| Total FA (mg/g) | 27.2±2.5 | 60.7±4.3[*] | 53.3±7.2[*] |

Values are presented as mean±SD (n=6/group). [*]$p < 0.05$ vs. Ctr. Abbreviations: FA, fatty acid; Ctr, control group; Chol, cholesterol group; Chol+Statin, cholesterol+lovastatin group; tr., trace amount (detected but below the limit of quantification); SD, standard deviation; SHRSP, stroke-prone spontaneously hypertensive rats.

## Discussion

Chol is essential for membrane integrity, and its low levels in SHRSP rats are linked to hemorrhagic stroke susceptibility [24]. While 1% Chol supplementation has been reported to extend lifespan in this model [8], the underlying mechanisms remain to be fully elucidated. In the present study, 12 weeks of Chol intake reduced SBP (Fig 2), consistent with prior reports of prolonged lifespan in this model [8]. This change was accompanied by a reduction in renal PhyS (Fig 3C). While the exact pathways were not directly assessed, it is possible that these lower tissue levels reflect a combination of competitive inhibition of intestinal absorption and hepatic sequestration, which might limit the release of PhyS into systemic circulation [12,25]. This interpretation is consistent with the observed hepatic sterol accumulation, although the model's inherent *Abcg5* mutation likely complicates sterol flux. Furthermore, preliminary fecal analysis (n=2) provided only a tentative hint of sterol-specific excretion differences (e.g., Sito vs. Camp); due to the minimal sample size and lack of statistical significance, these observations remain purely speculative (S1 Table). Collectively, these findings suggest that dietary Chol may be associated with SBP modulation through altered sterol homeostasis, though the specific molecular contributions remain a subject for future investigation. Beyond their structural role, Sito and Camp may act as ligands for LXRα [26]. Their accumulation in the liver (Fig 3A) might therefore be associated with altered LXRα signaling and subsequent changes in hepatic FA content (Table 5). While the potential formation of oxidized PhyS (OxiPhyS) in vivo could also contribute to such metabolic shifts [27,28], these mechanisms remain speculative as neither LXRα activity nor OxiPhyS levels were directly assessed in this study. Further investigation is required to define the precise molecular basis for these observations and their contribution to the FA profiles observed in this model. The distinct side-chain structures of Sito and Camp influence their micellar solubility and absorption [29,30]. While not directly characterized in SHRSP rats,

**Table 6. FA compositions and levels in serum of SHRSP rats.**

| FA (%) | Ctr | Chol | Chol+Statin |
|---|---|---|---|
| 14:0 | 0.6±0.1 | 0.5±0.1 | 0.6±0.1 |
| 16:0 | 22.7±0.9 | 21.0±0.8* | 21.3±0.5* |
| 16:1 | 1.2±0.2 | 2.4±0.3* | 2.3±0.9* |
| 18:0 | 12.4±0.9 | 9.5±0.9* | 9.0±1.4* |
| 18:1n-9 | 14.4±1.4 | 22.6±1.3* | 22.8±2.9* |
| 18:1n-7 | 2.0±0.2 | 2.7±0.1 | 2.4±0.5 |
| 18:2n-6 | 22.6±0.8 | 27.0±0.4* | 26.9±0.9* |
| 18:3n-6 | 0.3±0.0 | 0.2±0.0* | 0.2±0.0* |
| 18:3n-3 | 0.7±0.1 | 1.1±0.1* | 1.0±0.1* |
| 20:0 | 0.2±0.0 | 0.2±0.0 | 0.2±0.0* |
| 20:1n-9 | 0.2±0.0 | 0.3±0.0 | 0.2±0.0 |
| 20:2n-6 | tr. | tr. | tr. |
| 20:3n-9 | 0.2±0.0 | 0.3±0.0* | 0.3±0.1* |
| 20:3n-6 | tr. | tr. | tr. |
| 20:4n-6 | 18.0±1.9 | 8.6±1.2* | 9.1±2.4* |
| 20:5n-3 | 1.0±0.1 | 1.0±0.0 | 1.0±0.1 |
| 22:5n-3 | 0.7±0.1 | 0.6±0.1 | 0.6±0.2 |
| 22:6n-3 | 2.6±0.3 | 1.9±0.1* | 2.0±0.2* |
| Total FA (mg/mL) | 1.9±0.2 | 2.0±0.3 | 2.0±0.3 |

Values are presented as mean±SD (n=6/group). *$p < 0.05$ vs. Ctr. Abbreviations: FA, fatty acid; Ctr, control group; Chol, cholesterol group; Chol+Statin, cholesterol+lovastatin group; tr., trace amount (detected but below the limit of quantification); SD, standard deviation; SHRSP, stroke-prone spontaneously hypertensive rats.

human data suggest that Sito has lower absorption and higher hepatic clearance than Camp [25], which might parallel the differential accumulation observed in this study. Specifically, hepatic Camp levels increased while its levels in serum and kidney decreased (Fig 3A–3C), potentially reflecting preferential hepatic retention or altered trafficking of specific sterols. Given that different PhyS species can exert unique biological effects in vivo [31], the physiological significance of this distribution—particularly its impact on the disease phenotypes of the SHRSP model—remains to be clarified through further investigation. Regarding FA profiles, the proportion of AA significantly decreased in the serum and liver, while a similar downward trend was observed in the kidney (Tables 5–7). Crucially, the absolute AA content was significantly reduced in the liver and exhibited a clear, albeit non-significant, downward trend in the kidney. These findings suggest that the observed changes reflect a genuine depletion of the absolute AA pool in certain tissues rather than a mere shift in relative composition. Since AA is the essential precursor for eicosanoids—such as PGE$_2$, which regulates inflammatory responses —this reduction in the absolute availability of AA likely limits the substrate supply for prostanoid synthesis. This is consistent with the decreasing trend in hepatic and renal PGE$_2$ levels observed in this study (Table 5 and Figs 4, 5). Consistent with the importance of the AA pool, DHA administration has been reported to suppress hypertension in SHRSP rats by lowering absolute plasma AA levels [32]. Our results raise the possibility that a reduction in the absolute AA pool is a key factor associated with the physiological improvements observed in this model, even though the magnitude of this depletion varied across tissues. In our results, *Srebf1c* expression was significantly upregulated in both the Chol and Chol+Statin groups, whereas *Lxrα* mRNA levels remained unchanged (Fig 6B). This divergence indicates that *Srebf1c* regulation is not strictly mirrored by *Lxrα* transcript abundance in this model. Although unmeasured in this study, it is possible that the high dietary Chol intake increased the levels of oxidized Chol, which acts as a potent LXRα ligand

**Table 7. FA composition and levels of phospholipids in the kidney of SHRSP rats.**

| FA (%) | Ctr | Chol | Chol+Statin |
|---|---|---|---|
| 14:0 | 0.3±0.0 | 0.2±0.0 | 0.2±0.0 |
| 16:0 DMA | 2.3±0.1 | 2.2±0.1 | 2.3±0.1 |
| 16:0 | 26.1±1.7 | 24.7±0.5* | 24.9±0.3* |
| 16:1 | 0.4±0.1 | 0.6±0.2* | 0.4±0.0# |
| 18:0 DMA | 1.5±0.1 | 1.3±0.0* | 1.4±0.1* |
| 18:0 | 19.8±0.3 | 19.1±0.7 | 19.1±0.4 |
| 18:1n-9 | 7.9±0.3 | 8.7±0.5* | 8.7±0.4* |
| 18:1n-7 | 1.5±0.0 | 1.7±0.1* | 1.7±0.0* |
| 18:2n-6 | 7.5±0.3 | 10.9±0.6* | 10.6±0.3* |
| 18:3n-6 | tr. | tr. | tr. |
| 18:3n-3 | 0.3±0.0 | 0.3±0.0 | 0.3±0.1 |
| 20:0 | 0.2±0.0 | 0.2±0.0 | 0.2±0.0 |
| 20:1n-9 | tr. | tr. | tr. |
| 20:2n-6 | tr. | tr. | tr. |
| 20:3n-9 | tr. | 0.2±0.0* | 0.2±0.0* |
| 20:3n-6 | tr. | tr. | tr. |
| 20:4n-6 | 28.7±0.2 | 26.0±0.4 | 26.5±0.4 |
| 20:5n-3 | 0.4±0.0 | 0.6±0.2* | 0.5±0.0 |
| 22:0 | 0.5±0.0 | 0.4±0.0 | 0.5±0.1 |
| 22:5n-3 | 0.3±0.0 | 0.3±0.0 | 0.3±0.0 |
| 22:6n-3 | 2.0±0.2 | 2.2±0.6 | 1.8±0.1 |
| Total FA (mg/g) | 16.2±0.7 | 16.3±1.6 | 15.2±1.0 |

Values are presented as mean±SD (n = 5–6/group). *$p < 0.05$ vs. Ctr; #$p < 0.05$ vs. Chol. Abbreviations: FA, fatty acid; Ctr, control group; Chol, cholesterol group; Chol+Statin, cholesterol+lovastatin group; DMA, dimethyl acetal; tr., trace amount (detected but below the limit of quantification); SD, standard deviation; SHRSP, stroke-prone spontaneously hypertensive rats.

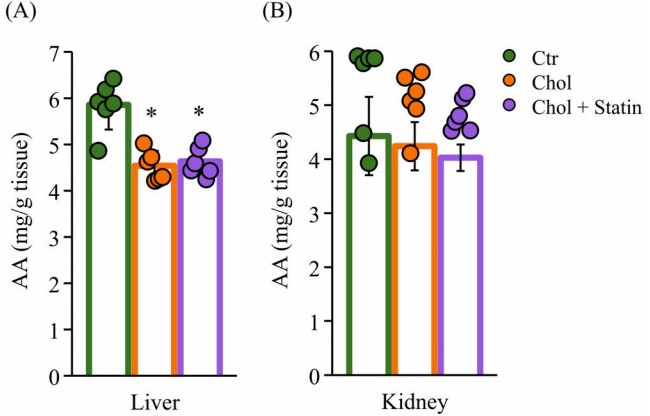

**Fig 4. AA levels in the liver and kidney of SHRSP rats. (A)** Hepatic AA levels in total lipids and **(B)** renal AA levels in the phospholipid (PL) fraction (mg/g tissue). Values are presented as mean±SD (n = 6/group). *$p < 0.05$ vs. Ctr. Abbreviations: Ctr, control group; Chol, cholesterol group; Chol+Statin, cholesterol+lovastatin group; SD, standard deviation; AA, arachidonic acid; SHRSP, stroke-prone spontaneously hypertensive rats.

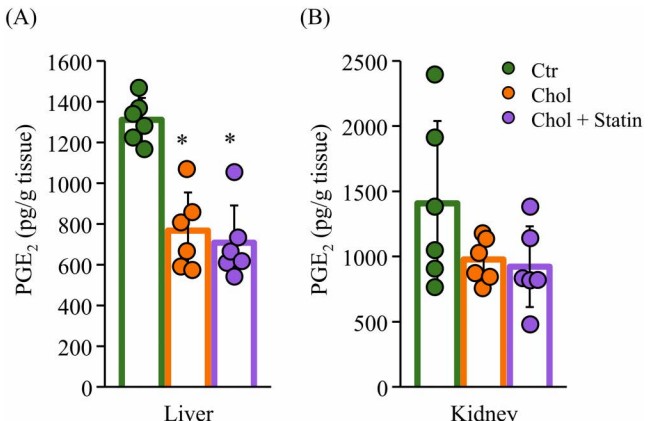

**Fig 5. PGE$_2$ levels in the liver and kidney of SHRSP rats. (A)** Hepatic and **(B)** renal PGE$_2$ levels (pg/g tissue). Values are presented as mean ± SD (n = 6/group). The PGE$_2$ concentrations were measured by ELISA using a kit (PGE$_2$ Monoclonal ELISA Kit). *$p < 0.05$ vs. Ctr. Abbreviations: Ctr, control group; Chol, cholesterol group; Chol + Statin, cholesterol + lovastatin group; SD, standard deviation; PGE$_2$, prostaglandin E$_2$; ELISA, enzyme-linked immunosorbent assay; SHRSP, stroke-prone spontaneously hypertensive rats.

[33]. Such ligand-mediated activation could explain the induction of *Srebf1c* even without an increase in *Lxrα* mRNA. However, this remains speculative, and alternative LXRα-independent mechanisms cannot be excluded. Regarding AA biosynthesis, *Elovl5* mRNA expression was significantly decreased in the Chol and Chol + Statin groups, whereas *Fads1/2* levels remained unchanged (Fig 6B). This suggests that while AA desaturation capacity was preserved, a bottleneck in biosynthetic flux (linoleic acid → AA conversion) may have occurred. Furthermore, since AA levels are also regulated by membrane remodeling processes—including the incorporation and release of FAs (the Lands' cycle)—alterations in these turnover dynamics could potentially contribute to the observed AA depletion beyond simple enzyme regulation [34]. However, as these specific pathways were not assessed, the relative contribution of synthesis versus remodeling remains to be determined. Collectively, these findings suggest that dietary Chol influences AA availability through complex metabolic shifts, likely involving *Elovl5* as a key regulatory node.

PGE$_2$ signaling is mediated through four receptors (EP1–4), with EP3 and EP4 generally associated with pro- and anti-inflammatory responses, respectively [35–37]. In this study, hepatic expression of *Cox-2*, *Ep3*, and *Ep4* was upregulated, with a notably greater increase in *Ep4* relative to *Ep3* (Fig 6C). This occurred in parallel with a decreasing trend in PGE$_2$ levels (Fig 5), suggesting a potential shift in the signaling balance toward anti-inflammatory pathways. Logically, the upregulation of *Cox-2* and EP transcripts may represent a compensatory feedback response to the reduced PGE$_2$ availability, which likely stems from the depletion of the absolute AA pool. While these findings hint at an altered inflammatory environment, the functional consequences remain speculative as receptor activity and COX-2 enzymatic protein levels were not directly assessed. Further investigation is needed to clarify how these molecular shifts influence the broader physiological outcomes observed in the SHRSP model. Although high oleic acid (OA) intake has been linked to reduced lifespan in SHRSP rats [5,6], OA levels were elevated in our tissues despite the use of a diet associated with longevity [8]. Notably, fecal FA composition (n = 2) did not differ between groups, suggesting that the observed changes in serum and tissue FA profiles resulted from altered internal metabolism rather than differences in absorption (S1 Table). Specifically, the elevation of OA is likely driven by the significant upregulation of *Srebf1c* and *Scd1* (Fig 6B), which enhances the biosynthetic capacity for monounsaturated fatty acids. Collectively, these findings indicate that disease-related phenotypes in this model are likely determined by the broader FA metabolic balance—including the depletion of AA and accumulation of OA—rather than the levels of a single FA species alone.

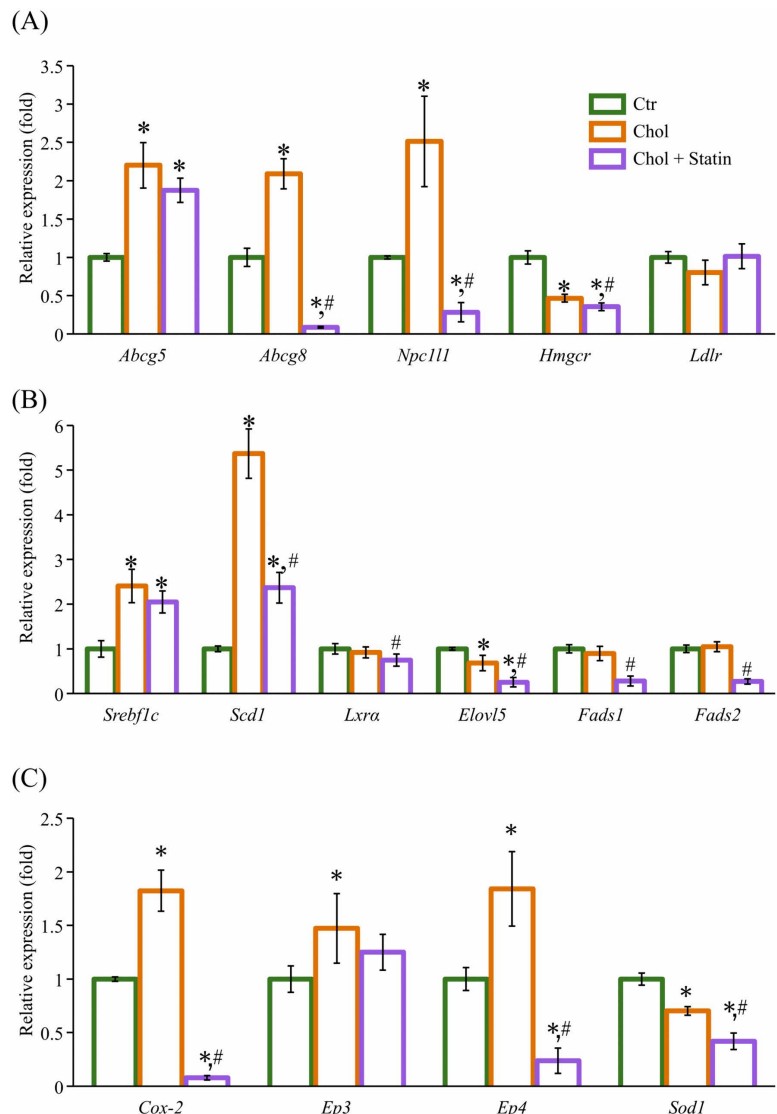

**Fig 6. Hepatic mRNA expression of lipid metabolism–related genes in SHRSP rats.** Hepatic mRNA expression related to **(A)** sterol metabolism, **(B)** FA metabolism, and **(C)** others. RNA samples from each group (n = 6) were pooled and each pooled RNA sample was analyzed in duplicate, and the experiment was independently repeated three times. Values are presented as mean ± SD of three independent assay runs using pooled RNA samples. The fold changes of target gene expression were determined with the ΔΔCt method, and the relative expression of these genes was calculated by the $2^{-\Delta\Delta CT}$ method and normalized to β-actin. Fold changes are presented on a linear scale. *$p < 0.05$ vs. Ctr; #$p < 0.05$ vs. Chol. Abbreviations: Ctr, control group; Chol, cholesterol group; Chol + Statin, cholesterol + lovastatin group; SD, standard deviation; *Abcg5*, ATP-binding cassette sub-family G member 5; *Abcg8*, ATP-binding cassette sub-family G member 8; *Npc1l1*, Niemann-Pick C1-like 1; *Hmgcr*, 3-hydroxy-3-methyl-glutaryl-coenzyme A (HMG-CoA) reductase; *Ldlr*, Low density lipoprotein receptor, *Srebf1c*, Sterol regulatory element-binding transcription factor 1c; *Scd1*, Stearoyl-CoA desaturase 1; *Lxrα*, Liver X receptor α; *Elovl5*, Elongation of very long chain fatty acids 5; *Fads1*, Fatty acid desaturase 1; *Fads2*, Fatty acid desaturase 2; *Cox-2*, Cyclooxygenase-2; *Ep3*, Prostaglandin E$_2$ receptor 3, *Ep4*, Prostaglandin E$_2$ receptor 4; *Sod1*, Superoxide dismutase 1; SHRSP, stroke-prone spontaneously hypertensive rats.

Regarding hepatic redox balance, dietary Chol was associated with decreased *Sod1* expression and SOD activity (S1 Fig). Although these changes might suggest a reduction in antioxidant capacity linked to hepatic lipid accumulation, the lack of direct oxidative stress measurements makes this interpretation speculative. The relationship between these

hepatic shifts and the systemic physiological changes observed in this model warrants further clarification in future studies. In contrast to reports linking rapeseed oil to decreased erythrocyte SOD activity in SHRSP rats [38,39], serum SOD activity was significantly increased in the Chol group (S1 Fig). While this may suggest an enhanced systemic antioxidant capacity, the functional implications of this change remain unclear as oxidative stress markers were not directly assessed. These results highlight the diverse physiological responses of this model to different dietary components and warrant further investigation.

Renal impairment in SHRSP rats is a known contributor to fatal stroke development [40]. In the present study, histological examination suggested that glomerular deformation and inflammatory cell infiltration were relatively milder in the Chol and Chol+Statin groups compared to the Ctr group (Fig 7 and Table 8). However, it must be emphasized that these histological differences were subtle—ranging only from "very slight" (±) to "slight" (+)—and were not supported by statistical analysis. While these modest findings may be associated with the lower SBP observed in the Chol-supplemented groups, a causal relationship cannot be established. In addition to morphological changes, alterations in the renal lipid environment might influence the activity of membrane-integrated functional proteins. For instance, dietary lipid modifications have been reported to alter renal lipid content and decrease Na,K-ATPase activity, possibly through changes in the membrane microenvironment or protein internalization [41]. This suggests that lipid-mediated regulation of ion transporters could be a contributing factor to the SBP modulation observed in our model, even when histological changes remain subtle. Furthermore, although alterations in renal sterol composition or FA metabolism might contribute to these pathological shifts, such mechanisms remain speculative as they were not directly assessed within the kidney. Overall, while dietary Chol appeared to be associated with a potential attenuation of renal damage under these conditions, further studies are required to validate the significance of these findings and their impact on long-term disease outcomes.

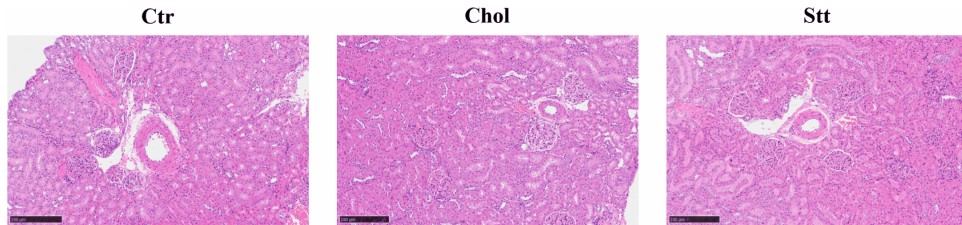

**Fig 7. Histological analysis of the kidney samples from SHRSP.** Tissue sections were stained with hematoxylin and eosin. Scale bar = 250 µm. Abbreviations: Ctr, control group; Chol, cholesterol group; Chol+Statin, cholesterol+lovastatin group; SHRSP, stroke-prone spontaneously hypertensive rats.

**Table 8. Summary of the histological analysis of the kidney from SHRSP.**

|  | Ctr | Chol | Chol+Statin |
|---|---|---|---|
| **Infiltrate, inflammatory cell, glomeruli** | (±)×3, (+)×3 | (±)×6 | (±)×6 |
| **Basophilia, tubule, cortex** | (±)×4, (+)×2 | (±)×5, (+)×1 | (±)×4, (+)×1 |
| **Cast, hyaline, tubule** |  | (±)×2 | (±)×1 |
| **Cast, granular, tubule** | (±)×1 | − | (±)×2 |
| **Infiltrate, inflammatory cell, interstitium** | (±)×5, (+)×1 | (±)×6 | (±)×6 |

Values indicate the number of rats in each histological grade. Histological changes were evaluated using a five-point scale (−, ±, +, ‡, ‡). Values represent the number of rats graded as ± or higher. Abbreviations: Ctr, control group; Chol, cholesterol group; Chol+Statin, cholesterol+lovastatin group; SHRSP, stroke-prone spontaneously hypertensive rats.

Statins act by inhibiting HMGCR and upregulating *Ldlr* expression to lower circulating Chol [42]. In this study, the Chol + Statin group exhibited decreased *Hmgcr* and a tendency toward increased *Ldlr* expression compared to the Chol group (Fig 6A), confirming that lovastatin actively modified hepatic Chol metabolism through its classic endogenous pathway even under high-dietary-Chol (1% w/w) conditions. Interestingly, although lovastatin further altered specific molecular signatures—including sterol profiles (Fig 3), serum SOD activity (S1 Fig), and FA-related gene expression (Fig 6B and 6C)—it produced no additional benefits in physiological or histological endpoints, such as blood pressure (Fig 2), renal AA content (Fig 4), PGE$_2$ levels (Fig 5), or glomerular morphology (Fig 7). These results suggest that while statin operates through a mechanistically distinct regulatory layer separate from dietary Chol, its molecular effects did not translate into detectable physiological differences under the present experimental conditions. Whether these statin-specific molecular shifts might influence long-term disease outcomes beyond the 12-week observation period remains to be determined.

A key strength of this study is the experimental design, which isolated the effects of dietary Chol from changes in FA intake—a common confounding factor in previous research. Furthermore, the use of SHRSP rats, with their unique genetic predisposition to altered sterol metabolism, provided a sensitive model to explore the complex interplay between Chol intake, lipid metabolism, and renal pathology. Several limitations remain. The high Chol dose and the specific Abcg5 mutation in SHRSP rats may limit the direct generalizability of these findings to humans or other models. Furthermore, as stroke incidence and specific signaling activities (e.g., COX-2 activity) were not assessed, the causal molecular mechanisms remain partly speculative. Future studies should employ dose-response analyses and longitudinal designs in various animal models to clarify the long-term impact of Chol and statins on cardiovascular health and lifespan.

## Conclusions

Dietary Chol (1% w/w) limits SBP elevation and renal damage in SHRSP rats by suppressing PhyS accumulation and the pro-inflammatory AA–PGE$_2$ signaling axis (Fig 8). Our findings, independent of dietary FA intake, indicate that these benefits stem from a fundamental shift in sterol homeostasis and a reduction in the absolute AA pool available for prostanoid synthesis. Collectively, this study offers novel insights into the potential protective roles of Chol in genetically susceptible individuals under stroke-prone conditions.

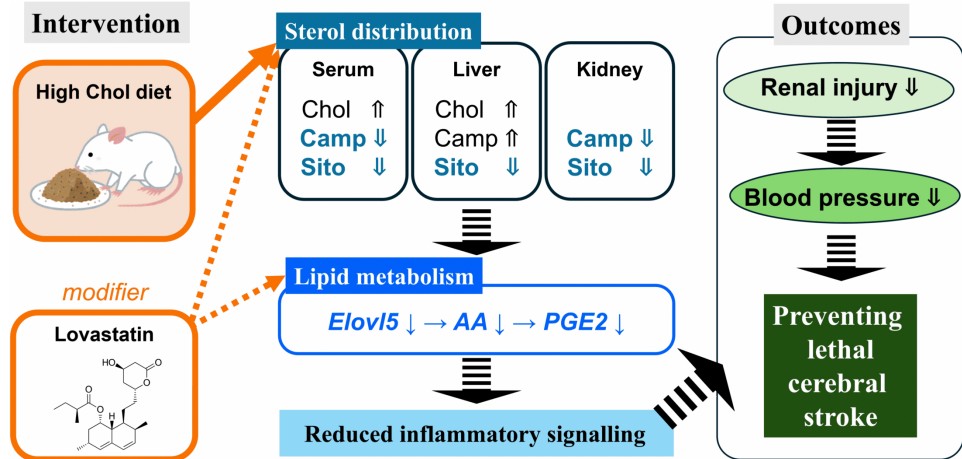

**Fig 8. Graphical summary of this study.** Dashed lines represent proposed mechanistic links and hypotheses. Abbreviations: Chol, cholesterol; PhyS, phytosterol.

## Supporting information

**S1 Table. FA compositions and levels, and sterol levels in the feces of SHRSP rats.** Each feces sample were collected from cages (n = 2/ group) at 16 weeks age (12-week feeding period). Abbreviations: Ctr, control group; Chol, cholesterol group; Chol + Statin, cholesterol + lovastatin group; SHRSP, stroke-prone spontaneously hypertensive rats.
(XLSX)

**S1 Fig. SOD activities in the liver, serum, and kidney of SHRSP rats.** (A) Hepatic, (B) serum, and (C) renal SOD activities (Unit/g tissue for liver and kidney; Unit/mL for serum). Each tissue homogenate were pooled in same dietary group, and each pooled RNA sample was analyzed in duplicate, and the experiment was independently repeated two times. Values are presented as mean (n = 2/group). OD activities were measured by colorimetry using a kit (SOD Assay Kit – WST). One unit (U) of SOD is defined as the amount of the enzyme that inhibits the reduction reaction of WST-1 with superoxide anion by 50%. $^*p < 0.05$ vs. Ctr; $^\#p < 0.05$ vs. Chol. Abbreviations: Ctr, control group; Chol, cholesterol group; Chol + Statin, cholesterol + lovastatin group; SOD, Superoxide dismutase; SHRSP, stroke-prone spontaneously hypertensive rats.
(TIF)

**S2 Table. Data of weight, food intake, organ weight, and systolic blood pressure.**
(XLSX)

**S3 Table. Data of FA analyses.**
(XLSX)

**S4 Table. Data of sterol analyses.**
(XLSX)

**S5 Table. Data of other biochemical analyses.**
(XLSX)

**S6 Table. Data of real-time RT-PCR.**
(XLSX)

## Acknowledgments

The authors thank S.N., who helps to maintain the animal experiments.

## Author contributions

**Conceptualization:** Kenjiro Tatematsu, Naoki Ohara.

**Data curation:** Yutaro Nishikata, Kenjiro Tatematsu.

**Formal analysis:** Yutaro Nishikata.

**Funding acquisition:** Kenjiro Tatematsu.

**Investigation:** Kenjiro Tatematsu, Yutaro Nishikata, Yoshiaki Saito.

**Methodology:** Yutaro Nishikata, Kenjiro Tatematsu, Yoshiaki Saito, Naoki Ohara.

**Project administration:** Yutaro Nishikata, Kenjiro Tatematsu, Naoki Ohara.

**Resources:** Kenjiro Tatematsu, Naoki Ohara.

**Software:** Yutaro Nishikata.

**Supervision:** Kenjiro Tatematsu, Yoshiaki Saito, Toshiyuki Matsunaga, Naoki Ohara.

**Validation:** Yutaro Nishikata, Kenjiro Tatematsu, Yoshiaki Saito, Toshiyuki Matsunaga, Naoki Ohara.

**Visualization:** Yutaro Nishikata, Kenjiro Tatematsu.

**Writing – original draft:** Yutaro Nishikata.

**Writing – review & editing:** Kenjiro Tatematsu, Yoshiaki Saito, Naoki Ohara.

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
