## [Decision Letter · Decision Letter 0]

5 Mar 2026

PONE-D-26-03912Dietary cholesterol reduces blood pressure and alters lipid profiles in stroke-prone spontaneously hypertensive ratsPLOS One

Dear Dr. Tatematsu,

Thank you for submitting your manuscript to PLOS ONE. After careful consideration, we feel that it has merit but does not fully meet PLOS ONE’s publication criteria as it currently stands. Therefore, we invite you to submit a revised version of the manuscript that addresses the points raised during the review process.

We look forward to receiving your revised manuscript.

Kind regards,

Luis Eduardo M Quintas, Ph.D.

Academic Editor

PLOS One

Journal Requirements:

“This work was supported by JSPS KAKENHI Grant Numbers JP22K05526.”

Reviewers' comments:

Reviewer's Responses to Questions

**Comments to the Author**

1. Is the manuscript technically sound, and do the data support the conclusions?

Reviewer #1: Yes

2. Has the statistical analysis been performed appropriately and rigorously? 

Reviewer #1: Yes

3. Have the authors made all data underlying the findings in their manuscript fully available?

Reviewer #1: Yes

4. Is the manuscript presented in an intelligible fashion and written in standard English?

Reviewer #1: Yes

5. Review Comments to the Author

Reviewer #1: Dear Editor of PLOS ONE,

Thank you for the opportunity to review manuscript PONE-D-26-03912, entitled “Dietary cholesterol reduces blood pressure and alters lipid profiles in stroke-prone spontaneously hypertensive rats.” The study contributes to a relevant discussion in nutrition and vascular pathophysiology by examining, in an experimental model of severe hypertension and hemorrhagic stroke, the apparent paradox between the traditionally assigned role of dietary cholesterol as a cardiovascular risk factor and prior evidence of increased survival in SHRSP animals fed a cholesterol-enriched diet.

Methodologically, the authors randomized male SHRSP/Izm rats (4 weeks old; n = 6/group) to three dietary conditions for 12 weeks: a control diet, a diet containing 1% (w/w) cholesterol, and a diet containing 1% cholesterol plus 0.025% (w/w) lovastatin (to suppress endogenous synthesis). They assessed pre/post systolic blood pressure and performed integrated analyses of sterols (including phytosterols), fatty acid composition, PGE₂ levels, and renal histopathology.

As the main findings, the cholesterol group exhibited a significant ~9–10% reduction in systolic blood pressure, accompanied by marked changes in the lipid profile, including modulation of phytosterols and a reduction in the arachidonic acid ratio in serum and kidney. In addition, the authors report a trend toward decreased PGE₂ and a modest attenuation of renal alterations. The authors interpret these results as evidence that dietary cholesterol may reduce renal phytosterol accumulation and suppress the AA–PGE₂ axis, potentially linking biochemical changes to improved renal/inflammatory status and to the phenotype previously associated with increased survival in this model.

Overall, I recommend a Major Revision. While the dataset is potentially suitable for publication in PLOS ONE, several issues should be addressed to ensure that the conclusions remain proportional to the evidence presented. In particular, sections of the Discussion and Conclusions repeatedly imply reduced hemorrhagic stroke risk and/or lifespan extension, yet survival and stroke incidence were not directly measured in the current study; these statements should be reframed as hypotheses grounded in prior reports and tied more explicitly to the measured endpoints (e.g., SBP changes, lipid profile shifts, and renal histology). In addition, there are internal consistency and reporting issues that require correction and greater transparency—most notably, a discrepancy regarding the direction of Ldlr expression in the statin group (as described in different sections), incomplete specification of statistical procedures (including distributional checks and handling of potential outliers), and figure presentation that would benefit from displaying individual data points given the small group sizes (n = 6). Addressing these points, along with careful moderation of mechanistic claims (e.g., inflammatory pathway activation inferred from mRNA and limited mediator measurements), would substantially strengthen the manuscript.

Minor/editorial comments

Introduction

L49–L51: The phrase “animal fats and oils” reads slightly imprecise (oils are typically plant-derived). Consider “animal-sourced foods” / “animal-derived foods,” depending on what you intend to emphasize.

L51–L52: “blood Chol” is broad. If you are referring to clinical/epidemiological evidence, please specify the metric (e.g., total cholesterol vs LDL-C), as this affects interpretation.

L52–L54: Consider replacing “widely acknowledged” with “well-established” to keep the tone neutral and improve readability.

L52–L54: The plaque sentence could be streamlined. For instance, instead of “primarily due to the propensity for the formation…,” consider a more direct construction such as: “because it promotes the development of cholesterol-rich plaques.”

L54–L56: In epidemiological writing, “inverse association” is often preferred over “inverse correlation” unless a correlation coefficient is being reported.

L59: There is a clear line-break/formatting issue (“…established as 59 a model…”). Please correct to ensure the sentence reads smoothly.

L60–L61: Consider tightening for flow and precision. For example: “In this strain, lifespan is curtailed by fatal strokes secondary to severe hypertension and renal impairment.”

L62 (abbreviation clarity—important): In lipid nomenclature, “PS” is widely used for phosphatidylserine. Because you use “PS” to mean phytosterols, I strongly recommend avoiding the abbreviation “PS” throughout the manuscript. Options: write “phytosterols” in full, or define an alternative (e.g., PhyS or plant sterols (PLS)). At minimum, add a note at first use (e.g., “phytosterols, hereafter abbreviated as PhyS to avoid confusion with phosphatidylserine”).

L62–L63: “shorten survival” (or “reduce survival”) is more idiomatic than “shorten the survival.” Also, “by increasing blood pressure” can be simplified to “by raising blood pressure.”

L62–L63: Since phytosterols are central to the paradox, consider briefly specifying whether the effect is attributed to phytosterols generally or to specific sterols (e.g., sitosterol/campesterol), depending on what the cited work supports.

L64–L66: The ABCG5 statement would benefit from a bit more specificity for a general readership: is the mutation functionally described as loss-of-function with impaired sterol efflux? Also, “target tissues” is vague—if supported by the cited literature, consider naming the most relevant tissues (e.g., kidney/vasculature/brain).

L66–L67: For readability, consider simplifying the causal chain. Example: “Accumulated phytosterols can partially replace membrane cholesterol, which may reduce membrane integrity and increase vascular fragility [10].”

L68–L69: Unless the cited work directly demonstrates rupture causality, consider a more cautious phrasing: “may predispose cerebral vessels to rupture, contributing to hemorrhagic stroke.”

L81: “blood pressure elevation [15–17]” appears as a sentence fragment (possibly due to formatting). Please revise so it connects grammatically to its subject/verb.

L81–L83 (abbreviation): “Stt” is not a standard abbreviation and is unclear at first read. If you mean statins, I suggest writing “statins” throughout (or use “lovastatin treatment” / “statin treatment” rather than “Stt”).

L82–L83: Consider tightening wording and making sure it reflects what the cited studies actually measured (survival vs stroke incidence/severity vs surrogate outcomes).

L83: “excessive dietary Chol intake” is subjective. Consider replacing with a concrete phrasing such as “a cholesterol-enriched diet” and/or specify the % (w/w) used.

L85–L86: Formatting issue in “sterol regulatory element86 binding proteins.” Please correct to “sterol regulatory element–binding proteins (SREBPs).”

L85–L86: Consider specifying which SREBP is most relevant here (SREBP-1 vs SREBP-2), depending on whether the focus is FA synthesis vs cholesterol homeostasis.

L87–L88: “have been identified as playing pivotal roles” can be simplified to a more neutral phrasing (e.g., “PUFAs… play key roles in inflammatory regulation”).

L88–L90: When introducing AA from PGs, you may optionally acknowledge AA also feeds other eicosanoid classes (TX/LT), but keeping it focused on PGE₂ is also reasonable if that is the main readout.

L90–L91: The bridge “Therefore, alterations in sterol composition may influence FA-derived inflammatory pathways” is important. Consider adding one short mechanistic clause (membrane remodeling, substrate availability, enzyme regulation) to make the link less implicit.

L91–L94: “remain to be elucidated” appears twice in close succession. Consider rephrasing one instance to “remain unclear” or splitting into shorter sentences.

L94–L97: Please avoid “Stt” unless clearly defined and used consistently. Also, because “PS” is commonly phosphatidylserine, I recommend the phytosterol abbreviation fix here as well.

L95–L97: “these interactions induce pathologies” is stronger than needed; consider “may contribute to disease-relevant pathways/phenotypes.” Likewise, “offers significant insights” could be toned down to “may provide insight” for a balanced tone.

Materials and Methods

L105–L106: Please provide a brief description of the basal diet (5001) macronutrient profile (or cite the datasheet), since background fat/sterol content can influence lipid outcomes. Also clarify whether 5001 was meal vs pelleted during feeding.

L106–L110: “Stt group” remains unclear/atypical. If this group is specifically Chol + lovastatin, consider renaming to “Chol+Lovastatin” (or “Chol+Statin”) and define it once.

L108–L110: For reproducibility, please describe how cholesterol and lovastatin were incorporated (powder vs solvent), mixing equipment/time, whether homogeneity was assessed, and whether diets were fed as meal or re-pelleted.

L112: “compositions of fatty FA and sterol contents” appears to be a typo. Suggest “fatty acid composition and sterol content.”

L207–L211: Please report whether normality (e.g., Shapiro–Wilk) and homogeneity of variances (e.g., Levene or Brown–Forsythe) were assessed before one-way ANOVA, and what was done if assumptions were violated.

L207–L211: Please clarify whether any outlier screening was performed (and which method), including pre-specified criteria and whether exclusions occurred. If none, a brief statement would improve transparency.

Results

Figures: With n = 6/group, I recommend plotting individual data points overlaid on bars/means (or switching to dot/box plots). Please ensure legends state exact n, whether n refers to animals vs cages (food intake), and whether error bars are SD vs SEM.

Where helpful (especially with small n), please consider also reporting the range (min–max) for key endpoints—either in figure legends, supplementary tables, or the Results text. This would make variability and group overlap easier to interpret alongside mean ± SD.

L216–L217: “throughout the 12-week feeding period” implies a repeated-measures analysis. If only endpoints were compared, please rephrase accordingly; if time-course statistics were used, specify the model.

L218: Typo: “ecrease” → “decrease.”

L220–L222 / Table 4: Please report organ weights normalized to body weight (e.g., liver/body weight ratio) and include body weight at sacrifice.

Table 4 (scope/rationale): Given this is a severe hypertension/stroke model, it would be helpful to clarify the rationale for the organ panel. If available, reporting heart, brain, and adipose tissue weights (absolute + normalized) could add interpretive value; if not collected, a brief justification would help.

L241–L242: Please specify whether sterols/FAs are reported as absolute concentrations (e.g., mg/g tissue; µg/mL serum) or relative composition (%). This affects how “increase/decrease” should be interpreted.

L243–L246: Please state key comparisons consistently (Ctr vs Chol vs Stt), then add Chol vs Stt where relevant. This will read more cleanly.

L246–L248: Grammar issue (“In relation to PS, resulted…”). Please revise and also remind the reader what the abbreviation refers to—ideally avoid “PS” given the phosphatidylserine conflict.

L247–L251: The phytosterol redistribution pattern is interesting (increase hepatic campesterol/total phytosterols; decrease serum and kidney). Consider adding one short mechanistic sentence (even if cautious) linking this to known transport/excretion pathways, or explicitly state that the mechanism remains unclear.

L249–L251: “β-sitosterol decreased across all tissues” should clarify whether this is absolute concentration or proportional composition (important given total phytosterols change in different directions by tissue).

L260–L267: Please clarify “total FA level” (absolute amount) vs “composition” (%). The AA decrease across tissues is a key observation; consider a short mechanistic hypothesis (LA→AA flux vs remodeling/Lands’ cycle), especially since AA decreases in Chol even when Fads1/2 do not change.

L264–L265: Define “kidney PL” operationally (phospholipid fraction). If liver/serum reflect different fractions, please make that explicit so readers compare appropriately.

Fig 4 / L287–L289: Please specify PGE₂ units and method (ELISA vs LC-MS/MS; normalization), and align wording with significance (liver significant vs kidney trend). Exact p-values would help.

Fig 5 / SOD: Please clarify how SOD activity was measured and normalized (tissue per mg protein; serum per mL; hemolysis checks). Also specify which statistical test was used for SOD (Methods currently notes it differed, but does not state how).

qPCR: Please ensure Methods/legends report housekeeping gene(s), ΔΔCt approach, primer sequences/efficiencies, biological and technical replicates, and how fold-changes are plotted (linear vs log).

L313–L320: Npc1l1 is often discussed in intestinal absorption; if hepatic Npc1l1 is central here, add a brief justification/citation. Also, the Ldlr description should explicitly indicate which pairwise comparisons are statistically significant (avoid qualitative terms without stats).

L337–L340: It would strengthen Results to explicitly connect increased Srebf1c/Scd1 in Chol to the MUFA shifts (Tables 5–6), even in one sentence.

Fig 6C vs Fig 4: You report increased Cox-2/Ep3/Ep4 mRNA in Chol while hepatic PGE₂ is decreased. This is not necessarily contradictory, but it deserves one brief clarifying sentence (e.g., substrate limitation due to AA reduction, post-transcriptional regulation, activity vs transcript).

Histology (Fig 7 / Table 8): Please clarify whether scoring was blinded, provide explicit grade definitions (±, +, etc.), indicate sections/fields examined per animal, and state whether ordinal data were analyzed statistically (or state explicitly if not analyzed due to low frequency).

Discussion

The Discussion is informative but feels overly long and somewhat repetitive, and several mechanistic chains are described with a level of certainty that goes beyond what was directly measured. I suggest tightening the Discussion by focusing on the primary observed endpoints (SBP, sterol redistribution, AA/PGE₂ patterns, renal histology) and presenting mechanistic links (e.g., LXR/oxysterols/OPS; systemic anti-inflammation; lifespan extension) more explicitly as hypotheses or future directions unless directly supported.

Consider adding a simple schematic/graphical summary (either as a main figure) that lays out your proposed working model: dietary cholesterol → changes in phytosterol handling/redistribution (liver vs kidney) + modulation of AA availability and PGE₂ → renal histology/BP phenotype, with the lovastatin arm shown as a modifier. A clear “working model” figure would make the narrative easier to follow and would help distinguish measured outcomes from inferred mechanisms.

Key questions for the authors

Question 1: In the epidemiological literature cited in the first paragraph, which cholesterol metric (total cholesterol vs LDL-C/HDL-C) and which hemorrhagic stroke subtype (intracerebral vs subarachnoid) are most responsible for the reported inverse association? How should readers think about that human framing relative to the SHRSP model used here?

Question 2: To make the “paradox” framing more convincing, could you clarify whether the previously reported lifespan extension on a cholesterol-rich diet reflects an effect of cholesterol per se, or whether broader diet features could have contributed (diet composition, phytosterol handling, or energy intake)? Were these factors monitored in the cited survival studies and/or in the present work??

Question 3: When you mention that “Stt” improves survival in SHRSP rats, do you mean statins as a class or a specific statin/dose? Since this study uses lovastatin, it would help if the Introduction matched the prior evidence (class vs specific compound) more explicitly to the intervention tested here.

Question 4: Were the three diets isocaloric and matched for macronutrient/fat content, aside from the added cholesterol and lovastatin? If not, please report energy density and explain how potential intake differences were handled.

Question 5: A key point for interpretation: are the AA decreases driven by reductions in absolute AA content, or are they mainly relative (% total FA) shifts due to expansion of other FA pools (e.g., MUFAs and/or 18:2n-6)? A clear statement on concentration vs composition (and the lipid fraction analyzed in each tissue) would help.

Question 6: Hepatic PGE₂ decreases while Cox-2 and EP receptor transcripts increase in the Chol group. How do you reconcile this pattern? Is your working model that reduced AA availability limits PGE₂ production despite higher Cox-2 mRNA, or were protein/activity measures considered to support pathway interpretation?

Question 7: Given the Abcg5 mutation, do you interpret increased hepatic Abcg5/8 and Npc1l1 mRNA as a compensatory response to sterol accumulation? If so, is there any functional readout (fecal/biliary sterols, protein levels, or another proxy) that supports altered sterol flux beyond transcript changes?

Question 8: I may have missed this in the Methods—was histological scoring performed blinded to group allocation, and were ordinal scores analyzed statistically (or explicitly treated as descriptive due to low frequency)? A short clarification here would increase confidence in the renal pathology interpretation.

Overall, I find the topic timely and the experimental approach interesting, particularly the attempt to isolate dietary cholesterol effects without changing fatty acid composition. However, several points require clarification and some conclusions should be toned down to better match the endpoints measured (SBP, lipid profiles, PGE₂, and histology, without direct survival/stroke outcomes). Addressing the questions above and improving reporting transparency (diet composition, statistics, and data presentation) would substantially strengthen the manuscript and its interpretability.

6. PLOS authors have the option to publish the peer review history of their article (what does this mean?). If published, this will include your full peer review and any attached files.

Reviewer #1: **Yes:** Israel Jose Pereira Garcia

---

## [Author Response · Author response to Decision Letter 1]

20 Apr 2026

Manuscript PONE-D-26-03921

Response to Reviewers

Dear Luis Eduardo M Quintas, Ph. D

Thank you for giving us the opportunity to submit a revised draft of our manuscript titled “Dietary cholesterol reduces blood pressure and alters lipid profiles in stroke-prone spontaneously hypertensive rats” to PLOS One. We would like to thank the editor and the reviewers for their thoughtful comments and constructive suggestions, which have helped us improve the quality of our manuscript. We have incorporated changes in response to the reviewers’ suggestions.

The major revisions in this version include the following:

1. We streamlined the Discussion section to make it more concise and focused on our primary findings.

2. We clarified the data presentation by correcting the statistical methods and plotting all individual data points to improve transparency.

We have also addressed all specific concerns raised by the reviewers, as detailed in the point-by-point responses below.

Comments from Reviewer #1

Introduction

Comment 1: L49–L51: The phrase “animal fats and oils” reads slightly imprecise (oils are typically plant-derived). Consider “animal-sourced foods” / “animal-derived foods,” depending on what you intend to emphasize.

Response: Thank you for pointing this out. The reviewer is correct, and we changed “animal fats and oils” to “animal-derived foods”.

Comment 2: L51–L52: “blood Chol” is broad. If you are referring to clinical/epidemiological evidence, please specify the metric (e.g., total cholesterol vs LDL-C), as this affects interpretation.

Response: We agree with this comment. Therefore, we distinguished total cholesterol from LDL cholesterol in the revised text.

Comment 3: L52–L54: Consider replacing “widely acknowledged” with “well-established” to keep the tone neutral and improve readability.

Response: Thank you for pointing this out. We changed “widely acknowledged” to “well-established”.

Comment 4: L52–L54: The plaque sentence could be streamlined. For instance, instead of “primarily due to the propensity for the formation…,” consider a more direct construction such as: “because it promotes the development of cholesterol-rich plaques.”

Response: Thank you for this suggestion. We agree. Therefore, we revised “, primarily due to the propensity for the formation of Chol-rich plaques” to “because it promotes the development of Chol-rich plaques”.

Comment 5: L54–L56: In epidemiological writing, “inverse association” is often preferred over “inverse correlation” unless a correlation coefficient is being reported.

Response: Thank you for pointing this out. We changed “inverse correlation” to “inverse association”.

Comment 6: L59: There is a clear line-break/formatting issue (“…established as 59 a model…”). Please correct to ensure the sentence reads smoothly.

Response: Thank you for pointing this out. We changed the line-break settings.

Comment 7: L60–L61: Consider tightening for flow and precision. For example: “In this strain, lifespan is curtailed by fatal strokes secondary to severe hypertension and renal impairment.”

Response: Thank you for this suggestion. We agree. Therefore, we changed “the lifespan is curtailed by fatal strokes that result from severe hypertension associated with renal impairment” to “lifespan is curtailed by fatal strokes secondary to severe hypertension and renal impairment”

Comment 8: L62 (abbreviation clarity—important): In lipid nomenclature, “PS” is widely used for phosphatidylserine. Because you use “PS” to mean phytosterols, I strongly recommend avoiding the abbreviation “PS” throughout the manuscript. Options: write “phytosterols” in full, or define an alternative (e.g., PhyS or plant sterols (PLS)). At minimum, add a note at first use (e.g., “phytosterols, hereafter abbreviated as PhyS to avoid confusion with phosphatidylserine”).

Response: Thank you for this suggestion. We agree with this, and we changed “PS” to “PhyS” throughout the manuscript.

Comment 9: L62–L63: “shorten survival” (or “reduce survival”) is more idiomatic than “shorten the survival.” Also, “by increasing blood pressure” can be simplified to “by raising blood pressure.”

Response: Thank you for pointing this out. We changed “shorten the survival” to “shorten survival” and “by increasing blood pressure” to “by raising blood pressure”

Comment 10: L62–L63: Since phytosterols are central to the paradox, consider briefly specifying whether the effect is attributed to phytosterols generally or to specific sterols (e.g., sitosterol/campesterol), depending on what the cited work supports.

Response: Thank you for pointing this out. We agree. Because the cited studies used phytosterols derived from vegetable oils (e.g., canola oil or soybean oil), which mainly contain sitosterol rather than purified individual sterols, we revised the sentence as follows:

“Diets rich in soybean- or rapeseed-derived phytosterols (PhyS), which mainly contain β-sitosterol (Sito), shorten survival by raising blood pressure [5–7].”

Comment 11: L64–L66: The ABCG5 statement would benefit from a bit more specificity for a general readership: is the mutation functionally described as loss-of-function with impaired sterol efflux? Also, “target tissues” is vague—if supported by the cited literature, consider naming the most relevant tissues (e.g., kidney/vasculature/brain).

Response: Thank you for pointing this out. We agree. According to the cited literature, the ABCG5 mutation is associated with the phytosterol accumulation, particularly in the vasculature, although it was not explicitly described as a loss-of-function mutation with impaired sterol efflux. Therefore, we revised the text to describe ABCG5 as “a sterol transporter involved in cholesterol efflux” and changed “target tissues” to “the vasculature.”

Comment 12: L66–L67: For readability, consider simplifying the causal chain. Example: “Accumulated phytosterols can partially replace membrane cholesterol, which may reduce membrane integrity and increase vascular fragility [10].”

Response: Thank you for your suggestion. We agree, and we changed “The accumulated PS has been shown to partially replace membrane Chol, thereby reducing membrane integrity and increasing vascular tissue fragility” to “Accumulated PhyS can partially replace membrane Chol, which may reduce membrane integrity and increase vascular fragility”

Comment 13: L68–L69: Unless the cited work directly demonstrates rupture causality, consider a more cautious phrasing: “may predispose cerebral vessels to rupture, contributing to hemorrhagic stroke.”

Response: Thank you for pointing this out. We agree. Since the cited work did not directly demonstrate rupture causality, we changed “are prone to rupture, resulting in hemorrhagic stroke” to “may predispose cerebral vessels to rupture, contributing to hemorrhagic stroke.”

Comment 14: L81: “blood pressure elevation [15–17]” appears as a sentence fragment (possibly due to formatting). Please revise so it connects grammatically to its subject/verb.

Response: Thank you for your suggestion. We revised the sentence for clarity. The revised text now states that “statins exert pleiotropic effects, including attenuation of oxidative stress, suppression of vascular inflammation, and improvement of blood pressure regulation.”

Comment 15: L81–L83 (abbreviation): “Stt” is not a standard abbreviation and is unclear at first read. If you mean statins, I suggest writing “statins” throughout (or use “lovastatin treatment” / “statin treatment” rather than “Stt”).

Response: Thank you for your suggestion. We agree. Therefore, we changed “Stt” to “statins” (which means a class) or “lovastatin” throughout the manuscript.

Comment 16: L82–L83: Consider tightening wording and making sure it reflects what the cited studies actually measured (survival vs stroke incidence/severity vs surrogate outcomes).

Response: We revised this sentence to more accurately reflect the cited studies. The text now states: “Although statins have been reported to extend survival and reduce stroke volume in SHRSP rats, their effects in the context of a cholesterol-enriched diet have not been fully examined.”

Comment 17: L83: “excessive dietary Chol intake” is subjective. Consider replacing with a concrete phrasing such as “a cholesterol-enriched diet” and/or specify the % (w/w) used.

Response: Thank you for pointing this out. We agree and we changed “under conditions of excessive dietary Chol intake” to “under conditions with a Chol-rich diet”

Comment 18: L85–L86: Formatting issue in “sterol regulatory element86 binding proteins.” Please correct to “sterol regulatory element–binding proteins (SREBPs).”

Response: Thank you for pointing this out. We corrected the formatting issue.

Comment 19: L85–L86: Consider specifying which SREBP is most relevant here (SREBP-1 vs SREBP-2), depending on whether the focus is FA synthesis vs cholesterol homeostasis.

Response: Thank you for your suggestion. We agree. Since we mainly focused on the FA synthesis in this study, we changed “SREBPs” to “SREBP-1c”.

Comment 20: L87–L88: “have been identified as playing pivotal roles” can be simplified to a more neutral phrasing (e.g., “PUFAs… play key roles in inflammatory regulation”).

Response: Thank you for pointing this out. We agree. We changed “have been identified as playing pivotal roles” to “play key roles”.

Comment 21: L88–L90: When introducing AA from PGs, you may optionally acknowledge AA also feeds other eicosanoid classes (TX/LT), but keeping it focused on PGE₂ is also reasonable if that is the main readout.

Response: Thank you for this suggestion. We agree that arachidonic acid also serves as a precursor for other eicosanoids, such as thromboxanes and leukotrienes. However, because PGE2 was the only eicosanoid directly measured in this study, we chose to keep the text focused and retained the broader phrase “other compounds.”

Comment 22: L90–L91: The bridge “Therefore, alterations in sterol composition may influence FA-derived inflammatory pathways” is important. Consider adding one short mechanistic clause (membrane remodeling, substrate availability, enzyme regulation) to make the link less implicit.

Response: Thank you for this suggestion. We agree. Therefore, we revised as below:

Therefore, alterations in sterol composition may exert an influence on FA-derived inflammatory pathways, potentially by altering the substrate availability through the regulation of FA metabolic enzymes.

Comment 23: L91–L94: “remain to be elucidated” appears twice in close succession. Consider rephrasing one instance to “remain unclear” or splitting into shorter sentences.

Response: Thank you for this suggestion. We agree, and we changed “remain to be elucidated” to “remain unclear” and “elucidating...” to “clarifying ...”.

Comment 24: L94–L97: Please avoid “Stt” unless clearly defined and used consistently. Also, because “PS” is commonly phosphatidylserine, I recommend the phytosterol abbreviation fix here as well.

Response: Thank you for this suggestion. We agree. Therefore, we changed “Stt” to “statins” and “PS” to “PhyS” throughout the manuscript.

Comment 25: L95–L97: “these interactions induce pathologies” is stronger than needed; consider “may contribute to disease-relevant pathways/phenotypes.” Likewise, “offers significant insights” could be toned down to “may provide insight” for a balanced tone.

Response: Thank you for pointing this out. We agree, and we changed “induce pathologies” to “may contribute to disease-relevant pathways” and “offers significant insights” to “may provide insights”.

Materials and Methods

Comment 26: L105–L106: Please provide a brief description of the basal diet (5001) macronutrient profile (or cite the datasheet), since background fat/sterol content can influence lipid outcomes. Also clarify whether 5001 was meal vs pelleted during feeding.

Response: Thank you for this suggestion. We have now added a brief description of the basal diet in the Materials and Methods section. The basal diet (LabDiet 5001, meal form) contains 24.1% protein, 5.1% fat (ether extract), 5.3% crude fiber, and 2.86 kcal/g metabolizable energy, according to the manufacturer’s datasheet.

Comment 27: L106–L110: “Stt group” remains unclear/atypical. If this group is specifically Chol + lovastatin, consider renaming to “Chol+Lovastatin” (or “Chol+Statin”) and define it once.

Response: Thank you for this suggestion. We agree. Therefore, we renamed “Stt group” to “Chol + Statin group”.

Comment 28: L108–L110: For reproducibility, please describe how cholesterol and lovastatin were incorporated (powder vs solvent), mixing equipment/time, whether homogeneity was assessed, and whether diets were fed as meal or re-pelleted.

Response: Thank you for pointing out an important issue. We added the information of cholesterol and lovastatin incorporation (powder), mixing equipment (PRO-GMS5) and mixing time (more than 30 minutes), assessment of diet homogeneity, and the fact that all diets were provided in meal form.

Comment 29: L112: “compositions of fatty FA and sterol contents” appears to be a typo. Suggest “fatty acid composition and sterol content.”

Response: Thank you for pointing this out. This was a typo. We changed “the compositions of fatty FA and sterol contents” to “the FA compositions and sterol contents”.

Comment 30: L207–L211: Please report whether normality (e.g., Shapiro–Wilk) and homogeneity of variances (e.g., Levene or Brown–Forsythe) were assessed before one-way ANOVA, and what was done if assumptions were violated.

Response: Thank you for pointing this out. According to your suggestion, we thought that parametric analyses are not applicable due to the small sample size (n = 6). Therefore, non-parametric analyses were performed. Group differences were assessed using the Kruskal–Wallis test, followed by the Steel–Dwass test for multiple comparisons.

Comment 31: L207–L211: Please clarify whether any outlier screening was performed (and which method), including pre-specified criteria and whether exclusions occurred. If none, a brief statement would improve transparency.

Response: Thank you for pointing this out. We agree, and we clarified this point as follows:

No outlier exclusion was performed, and all data points were included in the analysis.

Results

Comment 32: Figures: With n = 6/group, I recommend plotting individual data points overlaid on bars/means (or switching to dot/box plots). Please ensure legends state exact n, whether n refers to animals vs cages (food intake), and whether error bars are SD vs SEM.

Response: Thank you for pointing this out. We agree. Therefore, we plotted individual data points overlaid on bars. Also, we stated exact n, and error bars are SD.

Comment 33: Where helpful (especially with small n), please consider also reporting the range (min–max) for key endpoints—either in figure legends, supplementary tables, or the Results text. This would make variability and group overlap easier to interpret alongside mean ± SD.

Response: We appreciate this suggestion. In the revised manuscript, we have ensured that all individual data points are clearly plotted in the figures, which inherently shows the range (minimum and maximum) and the distribution of the data. Additionally, the raw numerical data for each individual are documented in the Supplemental Information (Table S3-7). We believe that this approach provides a more comprehensive view of our findings than simply reporting the min/max values in the text.

Comment 34: L216–L217: “throughout the 12-week feeding period” implies a repeated-measures analysis. If only endpoints were compared, please rephrase accordingly; if time-course statistics were used, specify the model.

Response: Thank you for pointing this out. We agree, and we rephrased the sentence as follows:

“Body weight and food intake were monitored weekly over the 12-week feeding period, and group differences were evaluated at each time point; no significant differences were observed.”

Comment 35: L218: Typo: “ecrease” → “decrease”.

Response: Thank you for pointing this out. We changed “ecrease” to “decrease”.

Comment 36: L220–L222 / Table 4: Pleas

---

## [Decision Letter · Decision Letter 1]

7 May 2026

Dietary cholesterol reduces blood pressure and alters lipid profiles in stroke-prone spontaneously hypertensive rats

PONE-D-26-03912R1

Dear Dr. Tatematsu,

We’re pleased to inform you that your manuscript has been judged scientifically suitable for publication and will be formally accepted for publication once it meets all outstanding technical requirements.

Kind regards,

Luis Eduardo M Quintas, Ph.D.

Academic Editor

PLOS One

Additional Editor Comments (optional):

Reviewers' comments:

Reviewer's Responses to Questions

**Comments to the Author**

1. If the authors have adequately addressed your comments raised in a previous round of review and you feel that this manuscript is now acceptable for publication, you may indicate that here to bypass the “Comments to the Author” section, enter your conflict of interest statement in the “Confidential to Editor” section, and submit your "Accept" recommendation.

Reviewer #1: All comments have been addressed

2. Is the manuscript technically sound, and do the data support the conclusions?

Reviewer #1: Yes

3. Has the statistical analysis been performed appropriately and rigorously? 

Reviewer #1: Yes

4. Have the authors made all data underlying the findings in their manuscript fully available?

Reviewer #1: Yes

5. Is the manuscript presented in an intelligible fashion and written in standard English?

Reviewer #1: Yes

6. Review Comments to the Author

Reviewer #1: (No Response)

7. PLOS authors have the option to publish the peer review history of their article (what does this mean?). If published, this will include your full peer review and any attached files.

Reviewer #1: **Yes:** Israel Jose Pereira Garcia

---

## [Editor Report · Acceptance letter]

PONE-D-26-03912R1

PLOS One

Dear Dr. Tatematsu,

I'm pleased to inform you that your manuscript has been deemed suitable for publication in PLOS One. Congratulations! Your manuscript is now being handed over to our production team.

Kind regards,

on behalf of

Dr. Luis Eduardo M Quintas

Academic Editor

PLOS One